# An Empirical Analysis on Quantization Schemes for Large Language Models

**Li Ou Hu (2024403397)**
Department of Quantum & Computer Engineering
Delft University of Technology, The Netherlands
Delft, 2628 CD
hlo24@mails.tsinghua.edu.cn

**Justinas Jučas (2024403399)**
Department of Quantum & Computer Engineering
Delft University of Technology, The Netherlands
Delft, 2628 CD
justinasjucas@gmail.com

**Eddy Yue (2024403429)**
School of Economics and Management
University of New South Wales
eddyyue123@gmail.com

37th Conference on Neural Information Processing Systems (NeurIPS 2023).

# 1 Background

Transformer based LLMs have shown to have impressive capabilities across various Natural Language Processing (NLP) tasks (Chen et al. [2023], Wang et al. [2023], Thirunavukarasu et al. [2023]). Such models, however, have significant computational overheads to their memory size. For example, the Llama Touvron et al. [2023] model can have 8 billion up to 405 billion parameters, with future models being even larger (Kaplan et al. [2020]). With a datatype of 16-bit floating point weights, such models can require 16 GB up to 810 GB of memory storage. Additionally, floating-point arithmetic operations require significantly more computational power than integer arithmetic. Therefore, weight quantization methods have been proposed (Deng et al. [2020]) for hardware deployment, which for example quantize the model weights down to 8-bit integers, or in the most extreme case, 1-bit weights. With quantization, the memory footprint of the LLM model is smaller, and the computational overhead is reduced which can have significant impact both for the speed and energy costs for very big models. However, this method comes at a cost of a drop in the model inference accuracy. Furthermore, additional scaling parameters $S, Z$ (elaborated in Section 2) have to be kept track of, in order to prevent exploding datatype sizes. Currently, there are several popular quantization schemes, namely Post Training Dynamic Quantization (PTDQ), Post Training Static Quantization (PTSQ), Quantization Aware Training (QAT).

# 2 Definition

The affine quantization scheme for `float32` to `int8` can for example be defined to be

$$x = x_q/S + Z \tag{1}$$

where $x$ is the original floating-point value, $x_q$ is the quantized `int8` value associated with $x$, $S$ is a positive `float32` scaling value, $Z$ is a `int8` zero-point, which represents 0 in the `float32` domain.

**Quantization methods:** We will briefly discuss the Post Training Quantization (PTQ) and Quantization Aware Training (QAT) quantization methods.

**During Training:** PTQ does not require any modification during training. The model is trained in standard floating-point precision, typically 32-bit, and the weights and activations are saved. In QAT, the model is trained with quantization effects simulated in each forward pass while retaining floating-point precision during the backward pass. This approach allows the model to adapt to quantization.

**During Inference:** In PTQ, the weights and activations are quantized after training. This can be done using static or dynamic quantization methods. For QAT, the backward pass uses the original floating-point weights $w$ and activations $a$ for gradient updates.

**Quantization Step** Each weight $w$ and activation $a$ is quantized as in Equation 1, where $S$ and $Z$ are parameters that adjust the range and offset for the integer representation.

**Dequantization Step** During inference, the quantized values are converted back to approximate floating-point values for computation, for example:

$$\text{result} = (S_w \cdot w_{\text{int8}} + Z_w) \cdot (S_a \cdot a_{\text{int8}} + Z_a) \tag{2}$$

Inference in QAT is the same as in PTQ, with quantized weights and activations used directly.

# 3 Related Work

Quantization of large language models is a well-established and evolving area of machine learning research. Numerous quantization algorithms are already used to optimize models for various cases[1]. However, research in quantization remains highly active with new, more efficient techniques being developed (for instance, Egiazarian et al. [2024]). Generally, quantization algorithms are divided in three categories:

---

[1] `https://huggingface.co/docs/transformers/v4.46.0/quantization/overview`

- **Post Training Dynamic Quantization (PTDQ).** The range (precision) of each activation is computed during the runtime. This approach provides positive results without much preprocessing effort, as it eliminates the need for a calibration dataset and does not involve model retraining or fine-tuning. However, it introduces runtime overhead due to the inference-time computation of activation ranges, which leads to slower performance (PyTorch [2024]).

- **Post Training Static Quantization (PTSQ).** The range for each activation is computed in advance at *quantization-time*, usually by passing representative data (calibration dataset) through the model and recording the activation values. It is an efficient and fast approach, however, very dependent on the choice of calibration dataset and is not flexible during the runtime (PyTorch [2024]).

- **Quantization Aware Training (QAT).** The range for each activation is computed at training-time: the loss of the training also depends on the degree of quantization. This is a potentially very efficient method, capable of reducing the weight sizes to several bits (i. e. more efficient than PTQ approaches), however, requires a lot of time to train, modifications to the training pipeline and usually significant amount of training data (Nagel et al. [2021]).

Currently, there exist several state-of-the-art PTQ algorithms used for LLM quantization, which we may use as the baseline for our research.

- **GPTQ** (Frantar et al. [2023]). The working principle is based on minimizing the quantization error using second-order optimization. GPTQ is extremely efficient: it can quantize GPT models with 175 billion parameters in approximately four GPU hours while maintaining reasonable accuracy

- **AQLM** (Egiazarian et al. [2024]). Achieves extreme compression by breaking down weight matrices into smaller components and optimizing them in a way that adapts to the input structure of the model. It achieves significant results when encoding weights using just 2-3 bits without sacrificing much performance.

Huang et al. [2024] have done an empirical study on the quantization of the state-of-the-art LLaMA3 model using different quantization schemes. They experimented with eleven Post-Training Quantization and two Quantization-Aware Training techniques. Their results showed that even though LLaMA3 still demonstrated superior performance after quantization, the performance degradation associated with quantization was significant and could lead to larger declines. PTQ techniques turned out to show better performance than the techniques that included retraining the model, likely due to insufficient re-training data.

## 4 Proposed Method

We propose to use the LLaMA 3.1 8B model for our initial model. First, we pick the baseline at the `float16` datatype, and evaluate its inference accuracy for different datasets and different PTQ and potentially ATQ techniques. Additionally, if possible, we evaluate the inference speed[2]. Then, given the obtained results of the different quantization techniques, we will experiment with the most optimal algorithms and modify them in a way that could potentially improve their performance on LLaMa 3.1 8B model. In particular, a promising idea is to combine PTSQ and PTDQ techniques, so that the model would be quantized both during *quantization-time* and also certain range adaptations could occur during the inference. In this way, the benefits of both approaches could be utilized.

We will compare our new approaches with several baseline approaches that correspond to the current state-of-the-art algorithms, such as GPTQ, AQLM, etc. The datasets that we propose to experiment on are similar to the ones that our initial model was trained on: publicly available text datasets such as WikiText[3], Common Crawl [4] and others. Last, in order optimize time, we will aim to utilize the existing codebase for conducting the experiments, for instance, the libaries that are presented in the Hugginface[5] website.

---

[2]This metric is heavily subject to the underlying hardware, the implementation of the computations on the hardware, and the measurement itself

[3]`https://huggingface.co/datasets/Salesforce/wikitext`

[4]`https://commoncrawl.org/`

[5]`https://huggingface.co/docs/transformers/main/main_classes/quantization`

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
