# OpenReview forum: "[Proposal-ML] An Empirical Analysis on Quantization Schemes for Large Language Models"
_tsinghua.edu.cn/THU/2024/Fall/AML — THU 2024 Fall AML Submission_

### Official Review · ~Un_Lok_Chen1 · 2024-11-08
**A Comparative Study on PTQ techniques**

**Rating:** 8
**Confidence:** 4

**Review:**

Summary:

In this project proposal, the authors aim to compare the different parameter quantization schemes for large language models, including SOTA PTQ algorithms such as GPTQ and AQLM. This comparative study may serve as a basis for optimizing for a novel quantization algorithm.

Pros:

1. The authors present the preliminaries on quantization schemes and introduce and compare different concepts such as PTQ and QAT to better prepare readers with background knowledge.

Cons:

A. Major issues

1. The logic flow of the proposal is somehow fragmented and recurrent. The structure of the proposal can be further improved by reorganizing the Definition and Related Work sections.

B. Minor issues

1. The main objective/contribution of this project could be summarized and highlighted in the first paragraph to give readers a clearer idea of the focus of this work.

2. Should the different quantization techniques be tested on multiple large models to demonstrate the robustness of the conclusion?

3. Does the project follow a similar workflow as the empirical study by Huang et al. mentioned in the Related Work section, or does it possess different objectives than previous work?

4. There are several typos and grammar mistakes, e.g. in the third line of the Proposed Method section, it should be “QTA techniques” rather than “ATQ techniques”.

---

### Official Review · ~Zihan_Yan2 · 2024-11-10
**A clear proposal with some minor issues**

**Rating:** 8
**Confidence:** 4

**Review:**

The proposal aims to conduct an empirical analysis of quantization schemes for large language models (LLMs). It plans to use the LLaMA 3.1 8B model as a starting point to evaluate the impact of different data types (such as float16) and various post-training quantization (PTQ) techniques, as well as potential quantization-aware training (QAT) methods, on model inference accuracy, also considering inference speed. The study will explore techniques that combine post-training static quantization (PTSQ) and post-training dynamic quantization (PTDQ) to optimize the model during both quantization and inference. The proposal has a clear structure, a well-defined problem, and a relatively detailed technical approach. Unfortunately, there are some minor formatting issues and a few grammatical mistakes in the proposal.

---

### Official Review · ~Tianxing_Yang1 · 2024-11-10
**Exploring the Combination of PTSQ and PTDQ in LLM Quantization**

**Rating:** 8
**Confidence:** 4

**Review:**

This proposal presents an exploration of combining PTSQ and PTDQ for quantizing large language models (LLMs).

**Pros:**
- The quantization of models holds significant importance for the development and practical application of current LLMs.
- The proposal includes a thorough discussion of related work and their respective methodologies.
- The proposed method section outlines an initial approach for combining these techniques.

**Cons:**
- The experimental section should not only evaluate inference speed but also compare the impact of different quantization methods on model performance. For instance, testing the same model with different quantization techniques on the MMLU dataset would provide more comprehensive insights. Ideally, the results should reflect a balance between model performance and inference speed.

**Minor Issue:**
- There is a typo in the phrase “can require 16 GB up to 810 GB of memory storage.”

---

### Official Review · ~Zhijie_shen3 · 2024-11-10
**LLaMa 3.1 8B model as empirical analysis on quantization schemes**

**Rating:** 8
**Confidence:** 4

**Review:**

**Summary**
The paper the paper investigates different quantization methods, including Post-Training Quantization (PTQ) and Quantization Aware Training (QAT). The primary goal is to reduce memory consumption and computational cost while minimizing the impact on model accuracy. And propose to use the LLaMA 3.1 8B model as empirical analysis.

### **Pros**
1. The authors provide a clear explanation of the quantization techniques, including PTDQ, PTSQ, and QAT. The detailed breakdown of each method shows a solid understanding of current quantization strategies and their trade-offs.

2. The propose to use of the LLaMA 3.1 8B model for empirical analysis is a strong choice, and the plan to evaluate different quantization methods on datasets like WikiText and Common Crawl demonstrates a thorough approach to benchmarking performance.

### **Suggestions**

1. Currently, the evaluation seems to focus primarily on memory reduction and inference speed. It would be beneficial to include additional metrics such as model accuracy, precision, recall, and F1-score post-quantization to provide a more comprehensive assessment of the impact on model performance.

2. The authors mention the possibility of accuracy degradation after quantization but do not provide detailed experiments or analysis on how different quantization schemes affect various NLP tasks. Conducting a deeper analysis on the accuracy trade-offs would strengthen the study.

---

### Official Review · ~Zihan_Lv1 · 2024-11-11
**Clear Research Focus and Justification**

**Rating:** 8
**Confidence:** 4

**Review:**

The combination of static and dynamic quantization methods is an innovative idea that could potentially yield better results in terms of model efficiency. The intention to test new modifications to these methods and adapt them to the LLaMA 3.1 8B model demonstrates creativity and an understanding of the evolving nature of quantization techniques. However, the level of innovation could be further enhanced by exploring more diverse techniques or highlighting how this method might address some of the unresolved challenges in existing quantization research.

---

### Official Review · ~Zheng_Jiang2 · 2024-11-11
**Thorough analysis of quantization schemes for LLMs**

**Rating:** 9
**Confidence:** 4

**Review:**

**Summary:** This paper presents an empirical analysis of quantization schemes for large language models (LLMs), focusing on the trade-offs between model accuracy and computational efficiency.

**Strengths:**
1. The paper provides a thorough comparison of different quantization schemes, offering insights into their respective advantages and disadvantages. This comprehensive analysis is valuable for researchers and practitioners looking to optimize LLMs for various applications.
2. The proposed combination of PTSQ and PTDQ is an innovative approach that could potentially offer the best of both worlds in terms of quantization.

**Weaknesses:**
1. The paper does not provide a detailed discussion on how quantization affects the performance of LLMs in various NLP tasks. This information is crucial for understanding the trade-offs involved.
2. There is limited discussion on the practical challenges of implementing the proposed quantization schemes, especially in terms of hardware requirements and software modifications.

---

### Official Review · ~Xin_Chen65 · 2024-11-11
**Well-structured proposal**

**Rating:** 8
**Confidence:** 4

**Review:**

The proposal presents a comprehensive study on the quantization of large language models (LLMs), which is a crucial area of research given the computational and memory demands of these models. It is methodologically sound and has the potential to contribute to the understanding and development of quantization techniques for LLMs.

strength: (1) The proposed method of using the LLaMA 3.1 8B model as a starting point and evaluating its inference accuracy with different quantization techniques is a solid approach. (2) The plan to experiment with the most optimal algorithms and potentially combine PTSQ and PTDQ techniques is innovative and could yield interesting results. (3) The related work section provides a good overview of the current state of quantization algorithms and their categorization.

weakness: (1) It could benefit from a discussion on how the findings might be generalized or scaled to other models or datasets beyond the initial LLaMA 3.1 8B model. (2) A brief mention of the resources required for the project would help in understanding the feasibility of the proposed experiments.

---

### Official Review · ~KAI_JUN_TEH1 · 2024-11-11
**A very good research direction**

**Rating:** 9
**Confidence:** 4

**Review:**

The proposal provides an explanation of the issues brought about by quantization and categorizes quantization methods, giving me a comprehensive understanding of quantization. In addition, the proposal presents two state-of-the-art (SOTA) works to help us grasp the current progress in quantization research. However, I’m curious about the author’s intention to combine two distinctly different types of quantization methods. This idea is innovative, but what is the motivation or basis for doing so? Moreover, I’m wondering whether your optimization goal is to accelerate inference or to improve accuracy. Based on my understanding, if you aim to cover all aspects, the amount of work involved would be unimaginable.

---

### Official Review · ~Eddy_Yue1 · 2024-11-12
**Practical Project Investigating Quantisation**

**Rating:** 9
**Confidence:** 4

**Review:**

Detailed proposal working towards making large language models more resource-efficient for practical applications. Additional clarity on how potential trade-offs, such as accuracy losses, will be handled would also enhance the proposal's feasibility.

---

### Official Review · ~Maanping_Shao1 · 2024-11-12

**Rating:** 8
**Confidence:** 3

**Review:**

The proposal "An Empirical Analysis on Quantization Schemes for Large Language Models" aims to evaluate various quantization techniques on the LLaMA 3.1 8B model to reduce computational and memory overhead without significantly impacting accuracy. The authors propose comparing techniques like Post-Training Dynamic Quantization (PTDQ) and Quantization Aware Training (QAT), with baselines such as GPTQ and AQLM. Their approach includes combining PTDQ and Post-Training Static Quantization (PTSQ) to balance memory efficiency with inference speed. This study is promising for improving the deployability of large language models in resource-constrained environments.

---

### Official Review · ~Zhu_Zhang6 · 2024-11-12
**A good proposal for quantization schemes analysis**

**Rating:** 9
**Confidence:** 3

**Review:**

**Summary:**
This proposal presents an empirical analysis of quantization schemes for large language models (LLMs), specifically focusing on the LLaMA 3.1 model. The research aims to evaluate the effectiveness of post-training quantization (PTQ) techniques, such as Post Training Dynamic Quantization (PTDQ) and Post Training Static Quantization (PTSQ), alongside Quantization-Aware Training (QAT). The authors propose testing and comparing these techniques using standard datasets, including WikiText and Common Crawl, with a focus on inference accuracy and speed. The study seeks to identify optimal quantization methods to improve model performance and reduce computational overhead.

**Strengths:**
1. **Relevant and Timely Topic:** Quantization is critical for deploying large language models efficiently, addressing memory and computational challenges in practical applications.
2. **Clear Methodology:** The proposal clearly outlines quantization methods and evaluation metrics, providing a well-defined structure for experimental analysis.

**Weaknesses:**
1. **Limited Scope on New Methods:** The proposal mainly focuses on existing quantization techniques without suggesting innovative quantization algorithms or approaches.
2. **Evaluation Metrics Clarity:** While accuracy and speed are mentioned, additional metrics like energy efficiency or storage savings would provide a more comprehensive assessment of quantization effectiveness.

**Questions:**
1. How will the authors address the potential drop in accuracy that often accompanies quantization?
2. Are there plans to measure the energy efficiency gains from quantization, especially for applications on edge devices?